LETTER TO THE EDITOR

# Emergence of a Novel Genotype of Pigeon Deltacoronavirus Closely Related to Porcine Deltacoronavirus HKU15 and Sparrow Deltacoronavirus HKU17 in a Live Poultry Market in Shandong Province, China

Guo-Lin Wang,[a] Li-Bo Li,[b] Jin-Jin Chen,[a] Qing-Chuan Wang,[c] Run-Ze Ye,[a] Li-Ming Li,[b] Ka-Li Zhu,[a] Wen-Guo Jiang,[b] Shen Tian,[a] Li-Qun Fang[a]

aState Key Laboratory of Pathogen and Biosecurity, Beijing Institute of Microbiology and Epidemiology, Beijing, China
bJining Center for Disease Control and Prevention, Jining, China
cJining Municipal Government Hospital Department, Jining, China

Guo-Lin Wang and Li-Bo Li contributed equally to this study. Author order was determined both alphabetically and in order of increasing seniority.

**KEYWORDS** live poultry market, novel pigeon deltacoronavirus, phylogenetic analysis, porcine deltacoronavirus, sparrow deltacoronavirus

Human infections by viruses from *Coronaviridae* were limited to *Alphacoronavirus* (Alpha-CoV) and *Betacoronavirus* (Beta-CoV) (1, 2) until infections of the porcine deltacoronavirus (PDCoV) among children were reported in Haiti in 2021 (3). The PDCoV was first found in Hong Kong in 2009 (4) and can cause gastrointestinal symptoms in pigs (5). At present, the origin of PDCoV is still unknown, but phylogenetic analysis shows that it is most closely related to sparrow deltacoronavirus (SpDCoV), indicating that PDCoV has the ability of interspecies transmission (4). In this study, two pigeon deltacoronaviruses (PiDCoV) closely related to PDCoV and SpDCoV strains were identified in a live poultry market (LPM) in Shandong Province, China.

Between August 2022 and March 2023, a total of 106 samples were collected in a LPM in Jining City, Shandong Province, China (see Table S1 and Fig. S1 in the supplemental material). The samples were screened for coronavirus (CoV) by a seminested PCR methods, and the PCR products were further sequenced to confirm the positive results for CoV. We found that 61 (57.5%) samples were tested positive for *Gammacoronavirus* (Gamma-CoV), which can be detected in all sample types (Table S1). Only 2 (1.9%) samples from feces and drinking water of pigeons were found positive for *Deltacoronavirus* (Delta-CoV) (Table S1).

Phylogenetic analysis of the partial RNA-dependent RNA polymerase (RdRp) gene showed that the Gamma-CoVs identified in the study formed three distinct lineages with nucleotide identity of 81.4 to 88.2% (Fig. S2A). A total of 51 strains shared high sequence similarity and were closely related to the infectious bronchitis virus (IBV) found in Fujian Province, China, in 2020, with 97.6 to 99.8% nucleotide identities, and nine strains shared high sequence similarity and were closely related to the pigeon CoVs identified in Guangdong Province, China, in 2014, with 93.0 to 96.0% nucleotide identities (Fig. S2A). Additionally, one strain was in the same lineage as avian CoVs identified in China and Australia, with 93.9 to 96.9% nucleotide identities (Fig. S2A). Similarly, phylogenetic analysis of the partial RdRp gene showed that the two Delta-CoVs identified in the study clustered in the same lineage and were closely related to SpDCoV HKU17 identified in Hong Kong, in 2011, with 95.6 to 96.2% nucleotide identities (Fig. S2B).

The two Delta-CoV reverse transcriptase PCR (RT-PCR)-positive pigeon samples then underwent next-generation sequencing (NGS), and one nearly complete genomic sequence of PiDCoV-WS38 was obtained. The genome organization of the PiDCoV strain is similar to those of SpDCoV and PDCoV strains with the characteristic gene order 5'-replicase open

Address correspondence to Guo-Lin Wang, guolin_wang2019@163.com, or Li-Qun Fang, fang_lq@163.com.

The authors declare no conflict of interest.

reading frame 1ab (ORF1ab), spike (S), envelope (E), membrane (M), and nucleocapsid (N)-3′ (Fig. 1A). Similar to SpDCoV strains, the PiDCoV strain contains four nonstructural (NS) proteins (NS6, NS7a, NS7b, and NS7c), as shown in Fig. 1A. To confirm the results of NGS and get more sequence information of another PiDCoV strain, PiDCoV-WS31, we designed specific primers for whole sequences of S, E, M, N, and NS genes based on the genome of PiDCoV-WS38 (Table S2). We finally successfully obtained those gene sequences of PiDCoV-WS31 and found that the accuracy of NGS sequencing was 100%.

Then whole-genome nucleotide sequence comparison showed that PiDCoV-WS38 shares a higher identity with five SpDCoV strains (86.7 to 92.0%), six PDCoV strains (88.5 to 88.7%), and one Quail strain (84.5%), and it has low identities with other avian Delta-CoVs, such as HKU11-934 (72.9%), HKU12-600 (71.4%), HKU13-3514 (75.3%), and HKU19-6918 (58.2%) (Table S3 and Fig. S3). The amino acid identities of ADRP, 3CL$^{pro}$, RdRp, Hel, ExoN, NendoU, and O-MT replicase domains between PiDCoV-WS38 and other closely related Delta-CoVs are summarized in Table S4, and the phylogenetic analysis of theses domains are shown in Fig. S4. Results showed that the concatenated seven-replicase domains of the PiDCoV-WS38 strain had more than 95.0% amino acid identities to the members of species of *coronavirus HKU15* (Table S4), suggesting that PiDCoV-WS38 belongs to this species.

Further comparison of structural proteins showed that the two PiDCoV strains identified in this study shared higher identities (94.0%, 94.9%, 99.2%) to SpDCoV HKU17 (HKU17-6124; GenBank accession number NC_016992) than to four other SpDCoV strains (ISU690-4, ISU690-7, ISU42824, and ISU73347; GenBank accession numbers MG812375, MG812376, MG812377, and MG812378, respectively) (89.6 to 91.0%, 92.9 to 93.4%, 92.3 to 93.1%), three pig-origin PDCoV strains (HKU15-155, HKU15-CHN-Tianjin, and HKU15-USA-Arkansas; GenBank accession numbers NC_039208, KY065120, and KR150443 respectively) (92.5%, 92.9 to 93.4%, 96.4 to 97.2%), and three human-origin PDCoV strains (HKU15-Haiti-Human-0081-4, HKU15-Haiti-Human-0256-1, and HKU15-Haiti-Human-0329-4; GenBank accession numbers MW685622, MW685623, and MW685624, respectively) (92.5%, 93.4%, 96.4 to 96.8%) in their E, M, and N proteins, respectively (Table S5). In contrast, the two PiDCoV strains shared a significantly lower identity to SpDCoV HKU17 (49.7%) and other four SpDCoV strains (81.0 to 85.3%) than to three pig-origin PDCoV strains (87.9 to 88.2%) and three human-origin PDCoV strains (88.0 to 88.3%) in their S protein (Table S5). These findings were also confirmed by phylogenetic tree analysis based on amino-acid sequences in which the two PiDCoV strains clustered in the same lineage and were closely related to SpDCoV and PDCoV strains for E, M, and N proteins (Fig. 1B). However, phylogenetic analysis showed that the two PiDCoV strains were closely related to PDCoV strains but distinct from the SpDCoVs-HKU17 strain for the S protein (Fig. 1B). For the NS proteins (NS6, NS7a, NS7b, and NS7c), the two PiDCoV strains shared 100% amino-acid identity with each other and were closely related to SpDCoV strains in the tree (Table S6 and Fig. S5). Overall, different combinations of structural proteins between two PiDCoV strains and other members of the species *coronavirus HKU15* suggested that the PiDCoV represents a novel member in that species.

PDCoV was first identified in specimens from pigs in Hong Kong in 2009 (4), and it later appeared in several U.S. states (6). At present, the PDCoV has spread to Korea, the Chinese mainland, and Thailand (7–9). In 2021, PDCoV infection among children with acute febrile illness was reported in Haiti from plasma samples collected between May 2014 and December 2015 (3). Regarding the SpDCoV, the strain of sparrow HKU17-6214 was first reported in a surveillance study in Hong Kong (4), and then another four SpDCoV strains more closely related to PDCoV were identified in the United States in 2017 (10). In this study, PDCoV and SpDCoV-like strains were identified from pigeons for the first time, which expanded the host range of the related viruses. Further, the novel PiDCoV strains shared higher identities with PDCoV strains than SpDCoV strains in the S protein, which was crucial for cell entry (11). Regarding the findings of this study, perspective surveillance of Delta-CoVs in LPMs and pigeon farms in a large region is imperative. At the same time, more studies are needed on the infectivity, pathogenicity, and transmission ability of the novel PiDCoVs.

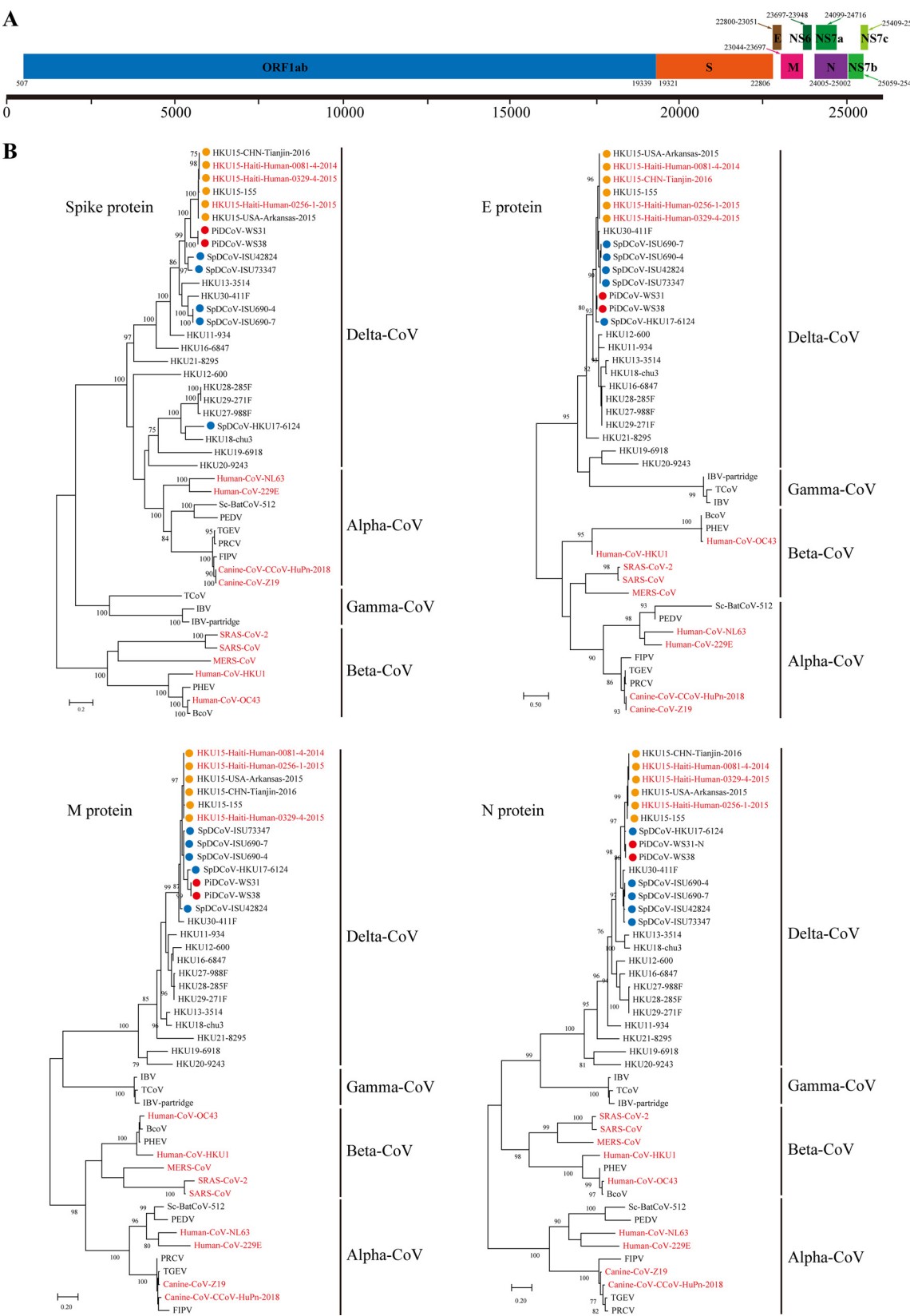

**FIG 1** Characterization of the pigeon deltacoronavirus (PiDCoV). (A) Genome annotation of PiDCoV-WS31. (B) Maximum-likelihood tree of spike (S), envelope (E), membrane (M), and nucleocapsid (N) proteins. The red, blue, and orange dotes represent PiDCoV, SpDCoV, and PDCoV strains, respectively. The strains with red font represent the virus that can infect humans.

**Data availability.** The data sets used and/or analyzed during this study are available from the corresponding author on reasonable request.

## SUPPLEMENTAL MATERIAL

Supplemental material is available online only.

**SUPPLEMENTAL FILE 1**, DOCX file, 1.8 MB.

## ACKNOWLEDGMENTS

This work was supported by the National Key Research and Development Program of China (number 2019YFC1200502 to Guo-Lin Wang), the National Natural Science Foundation (number 82103901 to Guo-Lin Wang), and the Beijing Natural Science Foundation (number L222119 to Guo-Lin Wang).

We declare no competing interests.

G.-L.W. and L.-Q.F. designed and supervised the research. L.-B.L., Q.-C.W., L.-M.L., and W.-G.J. collected samples and data. G.-L.W. and K.-L.Z. performed laboratory testing. G.-L.W, J.-J.C., R.-Z.Y., and S.T. analyzed the data. G.-L.W. and L.-Q.F. drafted the manuscript. All authors reviewed and approved the final manuscript.

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
