## [Reviewer comments · Microbiology Spectrum]

Microbiology Spectrum

The emergence of novel genotype of pigeon deltacoronaviruses closely related to porcine deltacoronaviruses HKU15 and sparrow deltacoronaviruses HKU17 in a live poultry market in Shandong province, China

Guo-Lin Wang, Li-Bo Li, Jin-Jin Chen, Qing-Chuan Wang, Run-Ze Ye, Li-Ming Li, Ka-Li Zhu, Wen-Guo Jiang, Shen Tian, and Liqun Fang

Corresponding Author(s): Guo-Lin Wang, Beijing Institute of Microbiology and Epidemiology

Review Timeline:

Submission Date:	February 5, 2023
Editorial Decision:	April 12, 2023
Revision Received:	May 19, 2023
Editorial Decision:	June 7, 2023
Revision Received:	June 8, 2023
Accepted:	June 9, 2023

Editor: Heba Mostafa

Reviewer(s): Disclosure of reviewer identity is with reference to reviewer comments included in decision letter(s). The following individuals involved in review of your submission have agreed to reveal their identity: Ganwu Li (Reviewer #2)

Transaction Report:

DOI: <https://doi.org/10.1128/spectrum.00556-23>

April 12, 2023

Dr. Guo-Lin Wang
Beijing Institute of Microbiology and Epidemiology
No. 20 Dongda Street, Fengtai District
Beijing 100071
China

Re: Spectrum00556-23 (The emergence of novel pigeon deltacoronaviruses closely related to porcine deltacoronaviruses and sparrow deltacoronaviruses in a live poultry market in Shandong province, China)

Dear Dr. Guo-Lin Wang:

Link Not Available

Sincerely,

Heba Mostafa

Journals Department
Reviewer comments:

Reviewer #1 (Comments for the Author):

The authors have done a virologic surveillance for coronaviruses from a small number of samples (n=50) from a poultry market in the Shandong Province of China. Using semi-nested RT-PCR followed by whole genome sequencing, they identified from 4% of the samples a pigeon deltacoronavirus which is closely related to sparrow deltacoronavirus HKU17, and the porcine deltacoronavirus HKU 15. However, their classification as novel species needs to be supported by the results from pairwise comparison of their 7 concatenated domains to those of known deltacoronaviruses based on the coronavirus species demarcation criteria by ICTV. This is not clearly spelt out in their results and the text.

Strength

1. This is an important topic because porcine deltacoronavirus related to porcine coronavirus HKU15 (which is closely related to sparrow coronavirus HKU17) has jumped into Haitian children.
2. The bioinformatic work is quite comprehensive

Weakness

1. Only one nearly complete genome of pigeon deltacoronavirus is sequenced.
2. The claim of a novel species has not been clearly demonstrated with the ICTV criteria.
3. The title would be more useful if they changed porcine deltacoronaviruses HKU15 and Sparrow Deltacoronaviruses HKU17 so that readers in this field can immediately grasp what the authors refer to.

Reviewer #2 (Comments for the Author):

Minor comments

1. From the Figure 1B, pigeon deltacoronaviruses also showed high nucleotide identities to human deltacoronaviruses from Haiti in the S gene sequences. It would be interesting to include the human deltacoronaviruses in the comparison from line 63 to line 76.
2. Line 64, please list strain names and GenBank accession numbers of four SpDCoV
3. Line 65, please list strain names and GenBank accession numbers of four PDCoV.

Staff Comments:

Preparing Revision Guidelines

Please return the manuscript within 60 days; if you cannot complete the modification within this time period, please contact me. If you do not wish to modify the manuscript and prefer to submit it to another journal, please notify me of your decision immediately so that the manuscript may be formally withdrawn from consideration by Microbiology Spectrum.

Responses to the reviewers' comments

Reviewer #1:

The authors have done a virologic surveillance for coronaviruses from a small number of samples (n=50) from a poultry market in the Shandong Province of China. Using semi-nested RT-PCR followed by whole genome sequencing, they identified from 4% of the samples a pigeon deltacoronavirus which is closely related to sparrow deltacoronavirus HKU17, and the porcine deltacoronavirus HKU 15. However, their classification as novel species needs to be supported by the results from pairwise comparison of their 7 concatenated domains to those of known deltacoronaviruses based on the coronavirus species demarcation criteria by ICTV. This is not clearly spelt out in their results and the text.

[Response] Thanks for your comments and we have re-clarified the pigeon deltacoronavirus as a novel member of species of *Coronavirus HKU15* by pairwise comparison of their 7 concatenated domains.

Strength

1. This is an important topic because porcine deltacoronavirus related to porcine coronavirus HKU15 (which is closely related to sparrow coronavirus HKU17) has jumped into Haitian children.
2. The bioinformatic work is quite comprehensive.

[Response] Thanks for your positive comments.

Weakness

1. Only one nearly complete genome of pigeon deltacoronavirus is sequenced.

[Response] Thanks for your comment. Between October 2022 and March 2023, we collected another 56 samples from the same live poultry market in Jining city of Shandong Province, China. However, all the new sequenced RdRp genes belonged to gammacoronavirus and the same viruses as previous pigeon deltacoronaviruses were not found (Lines 27-33). In the future, we will conduct a wider and longer-term surveillance around the area to continuously track

whether the pigeon deltacoronavirus will re-emerge.

2. The claim of a novel species has not been clearly demonstrated with the ICTV criteria.

[Response] Thanks for your comment. We have conducted the comparison of 7 concatenated domains between the pigeon deltacoronavirus and other closely related Delta-CoVs in the marked-up manuscript (Lines 63-70). We found that the pigeon deltacoronavirus belongs to the species of *Coronavirus HKU15*, however, the comparison of structural proteins suggested that the pigeon deltacoronavirus represents a novel member in that species (Lines 63-98).

3. The title would be more useful if they changed porcine deltacoronaviruses HKU15 and Sparrow Deltacoronaviruses HKU17 so that readers in this field can immediately grasp what the authors refer to.

[Response] Thanks for your comment and we have revised the title as suggested (Line 2).

Reviewer #2:

Minor comments

1. From the Figure 1B, pigeon deltacoronaviruses also showed high nucleotide identities to human deltacoronaviruses from Haiti in the S gene sequences. It would be interesting to include the human deltacoronaviruses in the comparison from line 63 to line 76.

[Response] Thanks for your useful suggestions. We have revised the relevant descriptions in the manuscript and the PDCoVs strains that were selected for comparison were divided into pig-origin PDCoV strains and human-origin PDCoV strains by the host origin (Lines 72-98).

2. Line 64, please list strain names and GenBank accession numbers of four SpDCoV.

[Response] Thanks for your suggestion. The strain names and GenBank accession

numbers of four SpDCoV strains has been listed in the marked-up manuscript (Lines 75-77).

3. Line 65, please list strain names and GenBank accession numbers of four PDCoV.
[Response] Thanks for your suggestion. The strain names and GenBank accession numbers of three pig-origin PDCoV and three human-origin PDCoV strains has been listed in the marked-up manuscript (Lines 77-83).

June 7, 2023

Dr. Guo-Lin Wang
Beijing Institute of Microbiology and Epidemiology
No. 20 Dongda Street, Fengtai District
Beijing 100071
China

Re: Spectrum00556-23R1 (The emergence of novel pigeon deltacoronaviruses closely related to porcine deltacoronaviruses HKU15 and sparrow deltacoronaviruses HKU17 in a live poultry market in Shandong province, China)

Dear Dr. Guo-Lin Wang:

Thank you for submitting your manuscript to Microbiology Spectrum. As you will see your paper is very close to acceptance. Please modify the manuscript along the lines I have recommended. As these revisions are quite minor, I expect that you should be able to turn in the revised paper in less than 30 days, if not sooner. If your manuscript was reviewed, you will find the reviewers' comments below.

When submitting the revised version of your paper, please provide (1) point-by-point responses to the issues raised by the reviewers as file type "Response to Reviewers," not in your cover letter, and (2) a PDF file that indicates the changes from the original submission (by highlighting or underlining the changes) as file type "Marked Up Manuscript - For Review Only". Please use this link to submit your revised manuscript. Detailed instructions on submitting your revised paper are below.

Link Not Available

Sincerely,

Heba Mostafa

Reviewer comments:

Reviewer #1 (Comments for the Author):

The paper is now satisfactorily revised.

JUST ONE important issue. Please change the title to "The emergence of novel genotype of pigeon deltacoronaviruses closely related to porcine deltacoronaviruses HKU15 and sparrow deltacoronaviruses HKU17 in a live poultry market in Shandong province, China

Reviewer #2 (Comments for the Author):

Revision is sufficient.

Preparing Revision Guidelines

Please return the manuscript within 60 days; if you cannot complete the modification within this time period, please contact me. If you do not wish to modify the manuscript and prefer to submit it to another journal, please notify me of your decision immediately so that the manuscript may be formally withdrawn from consideration by Microbiology Spectrum.

Responses to the reviewers' comments

Reviewer #1:

(1) The paper is now satisfactorily revised.

[Response] Thanks for your review and positive comment.

(2) JUST ONE important issue. Please change the title to "The emergence of novel genotype of pigeon deltacoronaviruses closely related to porcine deltacoronaviruses HKU15 and sparrow deltacoronaviruses HKU17 in a live poultry market in Shandong province, China

[Response] Thanks for your comment and the title has been revised as suggested.

Reviewer #2:

Revision is sufficient.

[Response] Thanks for your review and positive comment.

June 9, 2023

Dr. Guo-Lin Wang
Beijing Institute of Microbiology and Epidemiology
No. 20 Dongda Street, Fengtai District
Beijing 100071
China

Re: Spectrum00556-23R2 (The emergence of novel genotype of pigeon deltacoronaviruses closely related to porcine deltacoronaviruses HKU15 and sparrow deltacoronaviruses HKU17 in a live poultry market in Shandong province, China)

Dear Dr. Guo-Lin Wang:

Your manuscript has been accepted, and I am forwarding it to the ASM Journals Department for publication. You will be notified when your proofs are ready to be viewed.

Sincerely,

Heba Mostafa
Editor, Microbiology Spectrum
